# Evolutionary conservation and *in vitro* reconstitution of microsporidian iron–sulfur cluster biosynthesis

Sven-A. Freibert[1,*], Alina V. Goldberg[2,*], Christian Hacker[3,4], Sabine Molik[1], Paul Dean[2], Tom A. Williams[2], Sirintra Nakjang[2], Shaojun Long[2], Kacper Sendra[2], Eckhard Bill[5], Eva Heinz[2], Robert P. Hirt[2], John M. Lucocq[3], T. Martin Embley[2] & Roland Lill[1,6]

Microsporidians are obligate intracellular parasites that have minimized their genome content and sub-cellular structures by reductive evolution. Here, we demonstrate that cristae-deficient mitochondria (mitosomes) of *Trachipleistophora hominis* are the functional site of iron–sulfur cluster (ISC) assembly, which we suggest is the essential task of these organelles. Cell fractionation, fluorescence imaging and immunoelectron microscopy demonstrate that mitosomes contain a complete pathway for [2Fe–2S] cluster biosynthesis that we biochemically reconstituted using purified mitosomal ISC proteins. The *T. hominis* cytosolic iron–sulfur protein assembly (CIA) pathway includes the essential Cfd1–Nbp35 scaffold complex that assembles a [4Fe–4S] cluster as shown by spectroscopic methods *in vitro*. Phylogenetic analyses reveal that the ISC and CIA pathways are predominantly bacterial, but their cytosolic and nuclear target Fe/S proteins are mainly archaeal. This mixed evolutionary history of Fe/S-related proteins and pathways, and their strong conservation among highly reduced parasites, provides compelling evidence for the ancient chimeric ancestry of eukaryotes.

[1] Institut für Zytobiologie und Zytopathologie, Philipps-Universität, Robert-Koch-Strasse 6, Marburg 35032, Germany. [2] Institute for Cell and Molecular Biosciences, University of Newcastle, Newcastle upon Tyne NE2 4HH, UK. [3] School of Medicine, University of St Andrews, St. Andrews KY16 9TF, UK. [4] Bioimaging Centre, College of Life and Environmental Sciences, University of Exeter, Stocker Road, Exeter EX4 4QD, UK. [5] Max-Planck-Institut für Bioanorganische Chemie, Stiftstrasse 34-36, Mülheim an der Ruhr 45470, Germany. [6] LOEWE Zentrum für Synthetische Mikrobiologie SynMikro, Hans-Meerwein-Strasse, Marburg 35043, Germany. * These authors contributed equally to this work. Correspondence and requests for materials should be addressed to J.M.L. (email: jml7@st-andrews.ac.uk) or to T.M.E. (email: martin.embley@ncl.ac.uk) or to R.L. (lill@staff.uni-marburg.de).

Microsporidia are an enormously successful group of highly reduced obligate intracellular parasitic fungi that infect a wide range of eukaryotic hosts including immuno-compromised AIDS patients[1]. They are also pathogenic for a number of economically significant animals including key pollinators and fish[1]. Microsporidia have played a central role in hypotheses about early eukaryotic evolution and the importance of mitochondria in that process. These micro-organisms show a remarkable reduction of their metabolic capacities[2] as well as a striking simplification of core eukaryotic cellular structures including the mitochondrion[3]. This organelle appears as a tiny, double membrane-bounded cell compartment[3] commonly called a mitosome[4,5]. The discovery of the microsporidian mitosome[3] and similar organelles in, for example, Giardia[6] was an important milestone in formulating the hypothesis that mitochondria are an ancestral feature of all eukaryotic cells[5], but the selective pressure(s) governing their retention in modern eukaryotes are generally unknown, particularly for non-model species. Since some homologues of the mitochondrial iron–sulfur cluster (ISC) assembly machinery have been found encoded in microsporidian genomes and localized to mitosomes[4,7,8], it was suggested that this biosynthetic process was connected to the retention of the organelle. However, the biological function the microsporidian ISC proteins has not been experimentally verified, and the exclusive localization of some ISC components to the mitosome has been controversial[7].

The maturation of iron–sulfur (Fe/S) proteins is the only known essential biosynthetic function of Saccharomyces cerevisiae mitochondria[9]. The pathway is conserved in other eukaryotes including humans[10]. In brief, the process consists of three steps. First, a [2Fe–2S] cluster is synthesized on the scaffold protein Isu1 requiring the cysteine desulfurase complex Nfs1–Isd11 as a sulfur donor, the ferredoxin Yah1 as an electron donor and frataxin Yfh1 (ref. 11). Second, the newly synthesized cluster is released from Isu1 by a dedicated Hsp70 chaperone system (Ssq1, Jac1) and transferred to the monothiol glutaredoxin Grx5 from where [2Fe–2S] proteins can be assembled. The final step is specific for [4Fe–4S] proteins such as respiratory complexes I and II, and requires a dedicated set of ISC proteins including Isa1–Isa2–Iba57 for [4Fe–4S] cluster synthesis, and Nfu1, Bol1, Bol3 and Ind1 for specific cluster insertion into apoproteins[12,13].

Surprisingly, the essentiality of mitochondrial Fe/S protein biogenesis in yeast and humans is not explained by the maturation of endogenous, mitochondrial Fe/S proteins, but by the indispensable role of the ISC assembly machinery in the maturation of cytosolic-nuclear Fe/S proteins involved in key pathways of life including ribosome assembly and function, nuclear DNA replication and repair, chromosome segregation, and telomere length regulation (for review see ref. 14). The ISC assembly machinery produces an unknown, sulfur-containing compound that is exported by the mitochondrial ABC transporter Atm1 for utilization by components of the cytosolic iron–sulfur protein assembly (CIA) machinery that catalyse the maturation of both cytosolic and nuclear Fe/S proteins in various steps (reviewed in refs 15,16). First, a [4Fe–4S] is generated on the CIA scaffold complex Cfd1–Nbp35, which needs the Tah18–Dre2 electron transfer chain. The [4Fe–4S] cluster is then transferred to apoproteins via Nar1 and the CIA targeting complex Cia1–Cia2–Mms19. Maturation of the Fe/S protein Rli1 additionally and specifically requires the Yae1–Lto1 adapter complex.

To investigate the sub-cellular localization and functional importance of the microsporidian ISC components, we have used a combination of bioinformatics, cell biological and biochemical methods to show that Trachipleistophora hominis mitosomes contain a complete and functional pathway for the biosynthesis of [2Fe–2S] clusters similar to that recently documented for yeast mitochondria[11]. We have also identified the components of the microsporidian CIA pathway, and have functionally characterized the putative microsporidian CIA scaffold complex. Finally, by using phylogenetic methods we examined the evolutionary origins of the microsporidian ISC and CIA machineries as well as their cytosolic-nuclear target Fe/S proteins. Our work provides strong evidence for the functional importance of the mitosome for these highly reduced intracellular parasites in Fe/S protein metabolism. Further, the work allows detailed insights into the complex evolutionary history of this ancient and essential biosynthetic pathway of eukaryotes.

## Results

**Sub-cellular localization of microsporidian ISC components.** Hidden Markov models (HMM, see 'Methods' section) were used to search the genomes of two phylogenetically distinct microsporidians, Encephalitozoon cuniculi and T. hominis, for homologues of yeast and human ISC proteins. The two microsporidians differ in genome size and genomic content. E. cuniculi has a small genome of 2.9 Mb with ~2,000 genes[4], while the T. hominis genome is over 8.5 Mb with ~3,000 genes[8,17]. In addition to the previously detected five of ten core ISC homologues (that is, the cysteine desulfurase complex Nfs1–Isd11, the scaffold protein Isu1, frataxin Yfh1, a putative glutaredoxin Grx5 (ref. 7) and mitochondrial Hsp70 (Ssc1)[3,4]), we identified the genes for ferredoxin (Yah1), its reductase (Arh1) and the co-chaperone Jac1 in both genomes (Supplementary Table 1; Supplementary Fig. 1). Phylogenetic analyses recovered ThArh1 and EcArh1 in a monophyletic group with S. cerevisiae Arh1 (Supplementary Fig. 2), consistent with them being orthologues of the yeast mitochondrial enzyme. We did not detect a gene with similarity to the nucleotide exchange factor Mge1, which in yeast cooperates with Ssq1–Jac1 (ref. 18). Consistent with the absence of genes encoding mitochondrial [4Fe–4S] proteins such as aconitase and respiratory complexes I and II, we did not detect any ISC genes (for example, ISA1, ISA2 and IBA57;[19]) for this specific part of the ISC maturation pathway in either microsporidian suggesting that only the core ISC system for [2Fe–2S] protein assembly is present in microsporidia.

Previously, some microsporidian ISC proteins have been shown to localize to mitosomes in E. cuniculi and T. hominis using indirect immunofluorescence microscopy[7]. By contrast, T. hominis ThIsu1 and ThYfh1 were predominantly detected in the cytosol. To reinvestigate the functional locations of these two T. hominis core ISC components, we made new antibodies to T. hominis ThIsu1 and ThYfh1 for immunoelectron microscopy[3,20]. Glutaraldehyde-fixed, T. hominis-infected RK cells were cryo-sectioned and immunogold-labelled using these specific antisera. Both ThIsu1 and ThYfh1, like other core ISC proteins included as controls[3,7], localized exclusively to mitosomes that were detectable as small, double membrane-bounded organelles with minor and major axes ranging between 47 to 119 nm and 78 to 267 nm, respectively (Fig. 1a,b). To precisely localize the ISC proteins, we compared the distribution of gold label for each individual protein with that of random points, placed over the corresponding mitosome profiles (Fig. 1c,d; Supplementary Fig. 3). The resulting frequency distribution demonstrated that the ISC proteins were located predominantly towards the matrix side of the mitosomal inner membranes (86.8% in a 20 nm band inside the mitosome matrix and 13.2% outside). For independent

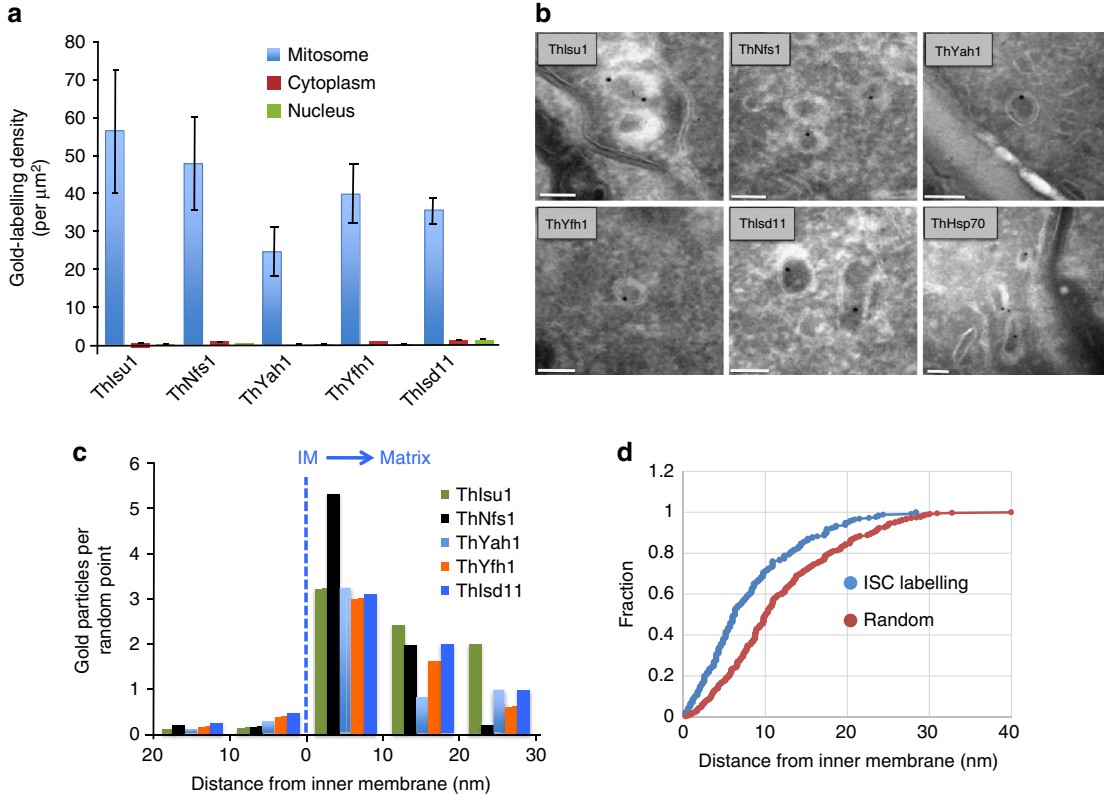

**Figure 1 | Sub-cellular localization of *T. hominis* Fe/S-cluster assembly components.** Thawed cryo-sections of glutaraldehyde-fixed *T. hominis*-infected RK-13 cells were labelled with antibodies to *T. hominis* ISC components and protein-A gold. (**a**) Quantitative analysis. Labelling is expressed as density of gold labelling over compartment profile area (estimated by point counting; see 'Methods' section; 34–40 micrographs analysed per protein/experiment with following number of golds per experiment: ThIsu1, 64; ThNfs1, 67; ThYah1, 35; ThYfh1, 70; ThIsd11, 111). Error bars indicate the s.e.m. (*N* = 3). (**b**) Images illustrating the distribution of labelling for ISC components in mitosomes (these show examples of positive labelling only and therefore do not reflect densities quantified in **a**). Labelling appears to be located over the matrix of the double membrane-bounded organelle profiles (with mean minor and major axes of 80 and 127 nm, respectively; *N* = 50 mitosome profiles). The analysed profiles were morphologically undistinguishable from mitosomes that labelled positively for ThHsp70 as shown here and described previously[3]. Bars = 100 nm. (**c**) The distribution of immunogold labelling for the indicated ISC proteins over mitosome profiles was compared with an equivalent number of random points. (IM, inner mitosomal membrane location). Labelling was expressed as gold particles per random point count and indicates the concentration of labelling at the IM/matrix interface. The numbers of gold and random counts were as follows: ThIsu1, 79; ThNfs1, 66; ThYah1, 37; ThYfh1, 63; ThIsd11, 50. Labelling for all tested ISC components was towards the mitosomal matrix with an enrichment at the IM. (**d**) Immunogold labelling distribution over mitosome matrix. Cumulative fraction plot shows pooled gold labelling data for all five ISC components (ISC labelling) compared with points located simple uniform random (random). ISC components show relative enrichment of labelling close to the IM (see Supplementary Fig. 3 for individual analyses).

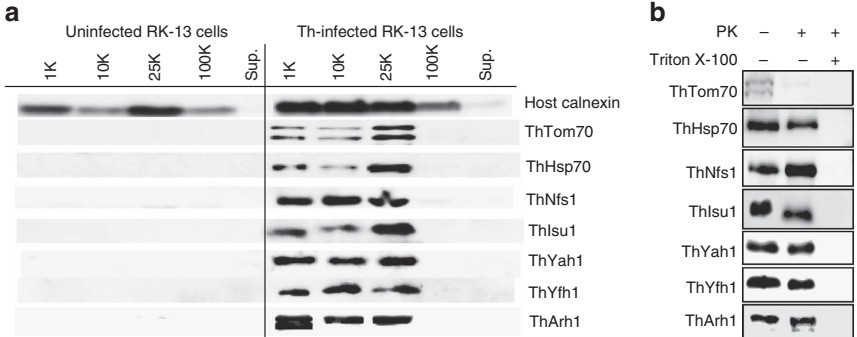

**Figure 2 | Biochemical localization of the *T. hominis* ISC pathway components.** (**a**) Western blots using antibodies to *T. hominis* proteins for fractions obtained by differential centrifugation from RK cells or RK cells infected with *T. hominis* (Th). Centrifugation speeds of pellet fractions are given above each lane. Sup = final 100,000*g* (100 K) supernatant. (**b**) Western blot of the 25,000*g* (25 K) pellet fraction of *T. hominis*-infected RK cells treated with or without proteinase K (PK) and Triton X-100 detergent as indicated.

biochemical verification of the mitosomal localization, mitosomes were enriched by differential centrifugation of *T. hominis*-infected cells. All analysed core ISC proteins (ThNfs1, ThIsu1, ThYah1, ThYfh1, ThArh1) were accumulated in the 25,000*g* (mitosome-enriched) pellet of infected RK-13 cells, which also contained the *T. hominis* homologue of mitochondrial

Hsp70 (ThHsp70), a validated mitosomal marker protein[3], and ThTom70, an orthologue of the yeast mitochondrial outer membrane protein import receptor Tom70 (Fig. 2a). The ISC proteins were protected against degradation by proteinase K, unlike the cytosol-exposed ThTom70, but were degraded on membrane lysis using detergent (Fig. 2b). These independent datasets support the exclusive mitosomal matrix location of all core ISC proteins.

**Functional analysis of the mitosomal ISC proteins.** Since there are no well-established genetic tools for the manipulation of microsporidians, we employed heterologous complementation of yeast mitochondrial ISC depletion mutants to test the functionality of the microsporidian ISC genes[7], although it is difficult to predict *a priori*, which genes will complement successfully. For example, targeting of *T. hominis* ThIsu1 to yeast mitochondria rescued growth of a conditional yeast mutant and restored mitochondrial Fe/S-cluster biosynthesis, but the *E. cuniculi* orthologue (EcIsu1) did not complement[7]. In the present study, we tested complementation of yeast ISC mutants depleted for Isd11, Arh1, Yah1 and Yfh1 by the respective *T. hominis* homologues. Correct mitochondrial localization of the *T. hominis* ISC proteins was achieved by fusing a well-characterized and highly effective fungal mitochondrial presequence to the N terminus of each ISC protein[7,21]. Of the four *T. hominis* ISC proteins tested (Supplementary Table 2) only ThIsd11 rescued growth of the respective Gal–ISD11 mutant (Supplementary Fig. 4a). Co-expression of ThIsd11 together with the cysteine desulfurase ThNfs1 actually inhibited the growth

of yeast wild-type cells suggesting a negative effect on endogenous cell function (Supplementary Fig. 4b). Complementation also failed when the electron transport chain partners ThArh1 and ThYah1 were co-expressed in the respective yeast mutants (Supplementary Table 2). The reasons why the tested microsporidian proteins did not complement yeast mutants, despite being unambiguous homologues of the respective yeast proteins, are not known. *T. hominis* ISC proteins are typically highly divergent in their primary sequence (see the long branches in the phylogenetic trees; Supplementary Fig. 2), and hence it is possible that they may not correctly interact with the residual endogenous yeast ISC partners[11]. However, the small number of residues known to be involved in yeast Nfs1–Isu1, Nfs1–Yfh1 and Isu1–Yfh1 interactions are conserved in *T. hominis* homologues suggesting that the mitosomal proteins still use similar interaction surfaces[22,23].

As a direct functionality test of the mitosomal ISC components we used a recently developed *in vitro* system that follows the *de novo* synthesis of a [2Fe–2S] cluster on the scaffold protein Isu1. Detection of this key reaction of mitochondrial Fe/S protein biogenesis makes use of a circular dichroism (CD) signal change at 431 nm that reflects Fe/S-cluster assembly on Isu1 (ref. 11). The *in vitro* reaction closely mimics the physiological situation, as it is kinetically fast, depends on all six ISC factors required *in vivo*, and occurs independently of the artificial reductant dithiothreitol. The *T. hominis* ISC proteins ThNfs1–ThIsd11, ThYfh1, ThYah1 and ThIsu1 were over-produced and purified from *E. coli* (Supplementary Fig. 5). Since we were unable to purify functional *T. hominis* ThArh1

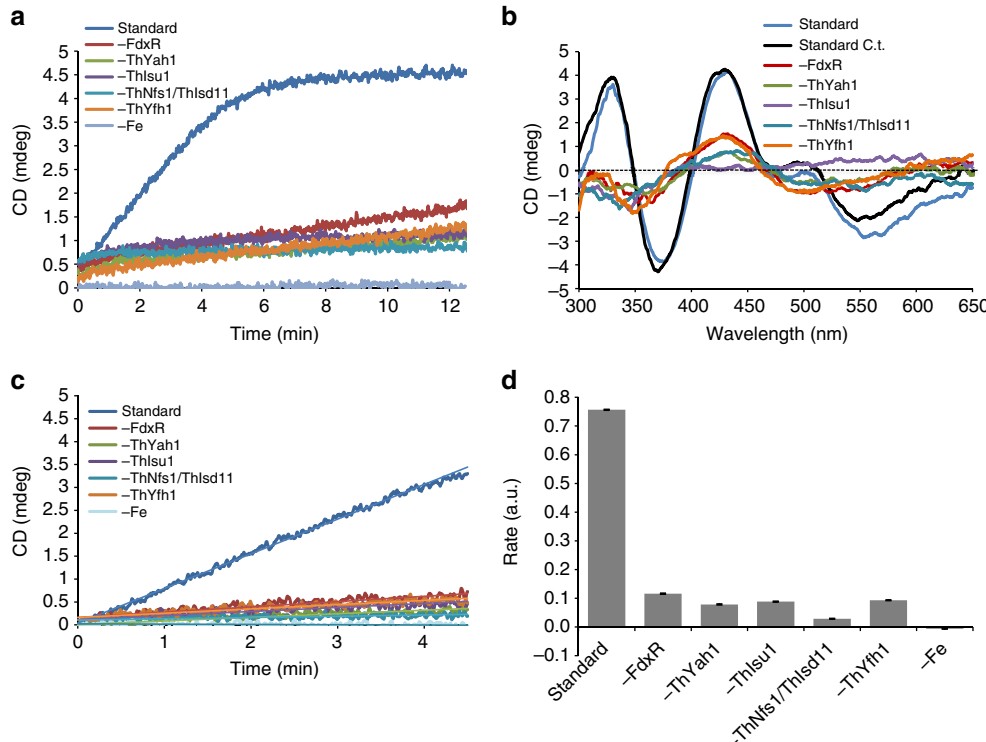

**Figure 3 | *In vitro* reconstitution of Fe/S-cluster synthesis on *T. hominis* Isu1.** (**a**) The *T. hominis* proteins ThIsu1, ThNfs–ThIsd11, ThYfh1, ThYah1 and human FdxR (the homologue of yeast Arh1) were mixed anaerobically in buffer R (standard reaction). Cysteine was added to start Fe/S-cluster synthesis, which was recorded by the CD signal change at 431 nm. Replicate reactions were performed, in which individual components of the *T. hominis* core ISC pathway or iron (Fe) were systematically omitted. (**b**) After 15 min full CD spectra were recorded for each reaction mixture, and compared with the spectrum of a standard reaction using ISC proteins from *C. thermophilum* (C.t.). (**c**) The initial rates of Fe/S-cluster synthesis on ThIsu1 for the different reactions were recorded. (**d**) The initial rates were estimated by linear regression and compared with the standard reaction using C.t. ISC components. $N \geq 5$; Error bars = s.d.

for this assay, we used purified human FdxR. This protein has previously been shown to be functional for *de novo* Fe/S-cluster synthesis with yeast ISC proteins[11]. The synthesis reactions were carried out anaerobically in the presence of reduced iron, cysteine and NADPH. When all purified *T. hominis* ISC proteins and human FdxR were mixed, a rapid and efficient generation of [2Fe–2S] clusters on ThIsu1

was observed (Fig. 3a,b). Omission of any of the six ISC factors greatly decreased the CD signal, as previously described in experiments using the yeast ISC proteins[11]. The initial rates of ISC factor-lacking reactions were close to the background levels observed in the absence of iron (Fig. 3c,d; see also ref. 11). Both the initial rates and the efficiencies of [2Fe–2S] cluster formation by the mitosomal ISC proteins were similar to those

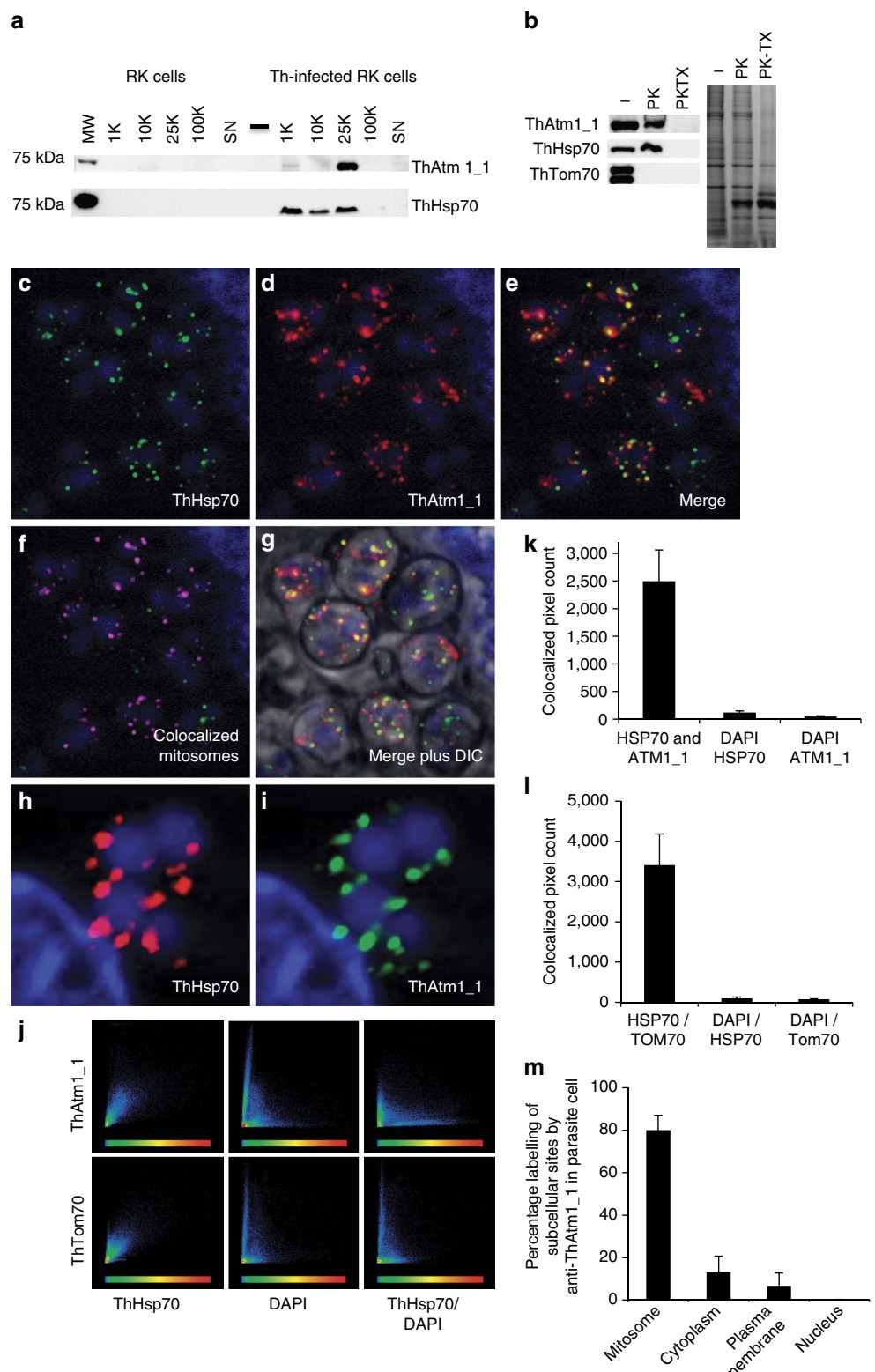

obtained with mitochondrial ISC proteins from either the thermophilic fungus *Chaetomium thermophilum* (Fig. 3b) or *S. cerevisiae*[11]. A comparison of the CD signal intensities of ISC-reconstituted and chemically reconstituted ThIsu1 (which binds one [2Fe–2S] cluster per dimer[11]) revealed the association of about one [2Fe–2S] per ThIsu1 dimer, similar to yeast or *Chaetomium* Isu1. The ability of mitosomal ISC components to *de novo* assemble a [2Fe–2S] cluster on ThIsu1 at rates and efficiencies comparable to those of mitochondrial ISC proteins demonstrates the functional equivalence of both biosynthetic systems.

**Localization and function of ISC export and CIA components.** We next investigated the components of the ISC export pathway[14]. The mitochondrial ABC transporter Atm1, together with glutathione (GSH) and Erv1, is crucial for yeast cytosolic and nuclear Fe/S protein biosynthesis, by exporting a sulfur-containing product that is essential for the function of the CIA machinery[24–27]. We first used an HMM-generated profile based on fungal Atm1 sequences to search for homologues in the predicted *T. hominis* proteome. Among 12 potential candidates, the three highest scoring *T. hominis* sequences (termed ThAtm1_1, 1_2 and 1_3) possessed structural features of the B sub-family of 'half-size' ABC transporters as expected for Atm1-like proteins (Supplementary Figs 1d and 6a)[28]. All three *T. hominis* sequences also clustered strongly with the yeast and *Neurospora* Atm1 protein sequences in phylogenetic analysis (Supplementary Figs 2 and 6b), consistent with their common ancestry.

Western blots and proteinase K protection assays of cell fractions identified a protein of the correct size for ThAtm1_1 in the mitosome-enriched fraction (Fig. 4), whereas no specific labelling was detected by immunostaining with antibodies against ThAtm1_2 or ThAtm1_3. Immunofluorescence microscopy using the antibody to ThAtm1_1 showed the co-localization of the punctate signals for ThAtm1_1 with those of the mitosomal marker ThHsp70 suggesting that ThAtm1_1 is a credible candidate for a functional homologue of yeast Atm1 (Fig. 4). We also tried to localize the ThAtm1 proteins by immuno-EM using specific antibodies, but did not observe specific labelling of *T. hominis* mitosomes or any other intracellular compartment. Further searches in the *T. hominis* or *E. cuniculi* genomes identified the two glutathione biosynthesis genes *GSH1* and *GSH2* (coding for γ-glutamyl-cysteine synthase Gsh1 and glutathione synthase Gsh2), but no orthologue of yeast *ERV1*, despite an earlier report of its presence in *E. cuniculi*[4]. On the basis of our analyses, we suggest that the gene reported in *E. cuniculi*[4] is a copy of the sulfhydryl oxidase Erv2, a paralogue of Erv1 located to the endoplasmic reticulum[29–31]. This gene was detected in both genomes.

We tested all three *T. hominis* Atm1 candidates for complementation of a yeast *ATM1* mutant using high and low level expression vectors with the addition of a fungal mitochondrial presequence[7,21] at the N terminus of the *T. hominis* sequences, but none of them showed any detectable rescue of the growth defect of an Atm1-deficient yeast mutant (Supplementary Fig. 7; Supplementary Table 2). Using this approach, we were thus unable to functionally confirm that ThAtm1_1 is a mitosomal homologue of yeast Atm1. However, in a recently published crystal structure of yeast Atm1 that contained a bound GSH ligand in a positively charged binding pocket[32], several residues were identified that are generally conserved in eukaryotes. These include the disease-relevant position D398 (corresponding to residue E433 in human ABCB7) in the case of X-linked sideroblastic anaemia and cerebellar ataxia (XLSA/A)[33], and yeast residues R280, R284 and N343. Intriguingly, in the putative *E. cuniculi* and *T. hominis* Atm1 sequences the latter residues are also conserved suggesting a functional similarity (Supplementary Fig. 1d).

We next searched the *T. hominis* and *E. cuniculi* genomes for homologues of components of the CIA pathway[14], using HMM models based on fungal sequences. We identified core CIA components including Tah18, Dre2, Cfd1, Nbp35, Nar1, Cia1 and Cia2 for both microsporidians, but did not find homologues of Mms19, Yae1 or Lto1 (Supplementary Table 1; Supplementary Fig. 1). To investigate the functionality of the *T. hominis* homologues, we tested their ability to rescue the growth of respective yeast CIA depletion mutants. None of the tested genes (*CFD1*, *NBP35*, *NAR1*, *CIA1*) showed any positive result (Fig. 5a; Supplementary Table 2). Co-expression of ThCfd1 and ThNbp35 also did not show complementation (Fig. 5a, right). We therefore employed an *in vitro* functional test for the ability of ThCfd1 and ThNbp35 to assemble a [4Fe–4S] cluster[34]. Both *E. cuniculi* and *T. hominis* Cfd1–Nbp35 contain the conserved cysteine residues shown to coordinate the Fe/S clusters of this complex[35] (Supplementary Figs 1e and 8a). ThCfd1 and His-tagged ThNbp35 were co-overexpressed in *E. coli* and co-purified indicating hetero-complex formation (Fig. 5b, inset right). The brownish colour of the protein solution indicated the presence of Fe/S clusters, but at under-stoichiometric amounts. We therefore chemically reconstituted ThCfd1–HisThNbp35 under anaerobic conditions in the presence of ferric ammonium citrate and Li$_2$S to yield a dark-brown protein complex with 9.5 Fe and 8.6S bound per hetero-dimer (Fig. 5b, inset left)[34]. This amount is close to the expected stoichiometry of 8 Fe and 8S for binding of the N terminal and bridging [4Fe–4S] clusters. Both ultraviolet–vis and electron paramagnetic resonance (EPR) spectroscopy showed spectral features (absorption peak at 420 nm; EPR *g* values of 1.89, 1.92 and 2.05) consistent with the presence of [4Fe–4S] clusters (Fig. 5b,c). When ThCfd1 was purified and reconstituted

**Figure 4 | Localization of the *T. hominis* homologue of mitochondrial Atm1.** (**a**) Fractionation by differential ultracentrifugation was carried out on healthy and *T. hominis*-infected RK cells and the fractions were immunostained using an antibody to ThAtm1_1. (**b**) Proteinase K (PK) protection was performed on the mitosome-enriched 25,000*g* (25 K) fraction of infected cells, with Triton X-100 (TX) used to solubilize the membranes. A Coomassie-stained gel of the fractions after the protection assay is shown on the right. (**c–i**) Immunofluorescence of fixed RK-13 cells infected with *T. hominis* using antibodies raised against ThHsp70 (**c**,**e**; green; rat) or ThAtm1_1 (**d**,**e**; red; rabbit). DAPI was used to label host and parasite nuclear DNA (blue). The ThAtm1_1 antiserum was pre-purified using uninfected RK-13 cell lysate immobilized on nitrocellulose following SDS–PAGE. Colocalized pixels were identified using Zeiss Axiovision software, and are pseudo-coloured pink as shown in **f**. The DIC image in (**g**) shows the individual meronts in the infected RK cell. (**h**,**i**) A close-up of a different sample of meronts showing the typical distribution of the signals for the two antibodies. (**j**) Colocalisation scatter plots against the different channels were generated by Axiovision software and the antibody to the protein import receptor ThTom70 was used as a positive control for colocalisation with ThHsp70. The scatter plots for ThAtm1_1 with ThHsp70 and ThTom70 with ThHsp70 are similar, indicating co-localization of both proteins with ThHsp70. (**k-m**), The colocalized pixel counts were also quantified (*n* = 3, error bars = s.d.) and represented graphically for ThHsp70 and either (**k**) ThAtm1_1 or (**l**) ThTom70. (**m**) The extent of mitosomal labelling by the ThAtm1_1 antibody relative to other cell compartments was quantified based on the proportion of pixel intensity across six fields of view for each of three replicates (error bars = s.d.).

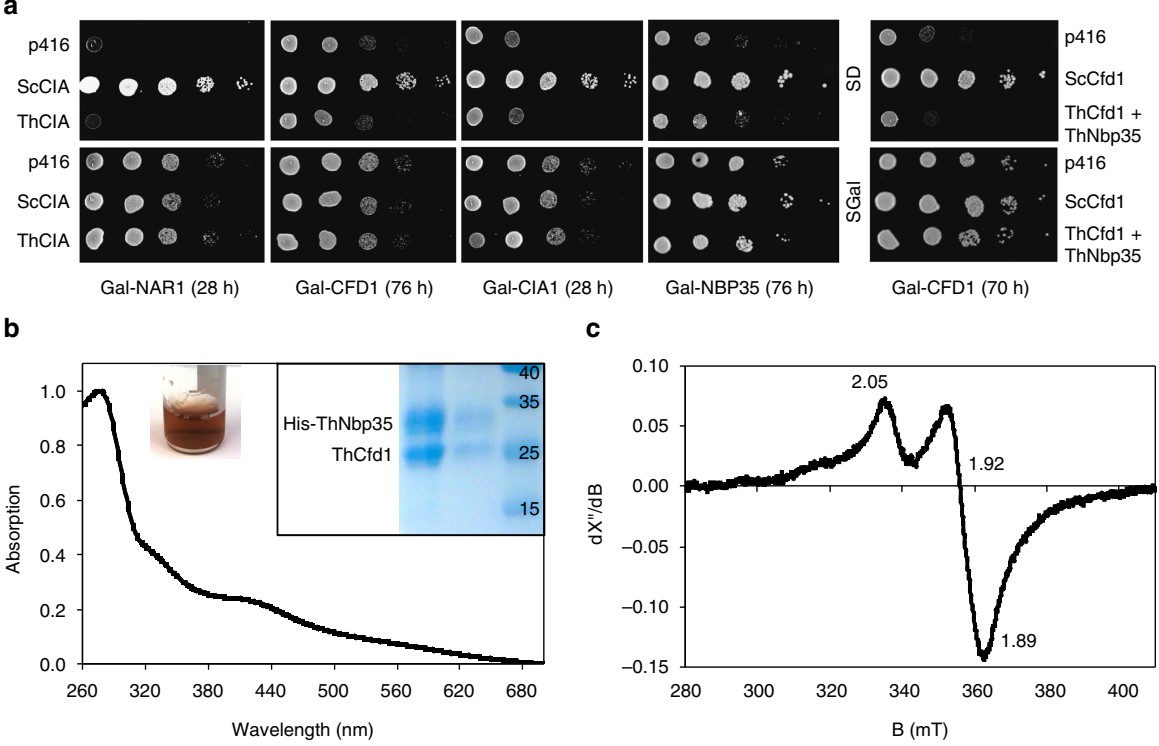

**Figure 5 | Functional reconstitution of the microsporidian CIA scaffold complex.** (**a**) Yeast complementation assay. Functionality of the *T. hominis* CIA components ThNar1, ThCfd1, ThCia1 or ThNbp35 was tested by their ability to rescue the growth defect of respective regulatable Gal-CIA yeast mutants on glucose-containing minimal medium (s.d.). Indicated cells were transformed with plasmid p416-MET25 containing either no gene (p416), the respective yeast (Sc) or the homologous *T. hominis* (Th) CIA genes. Gal-CFD1 cells were additionally transformed with p415-MET25 to allow co-production of both ThCfd1 and ThNbp35 (right). Cells were depleted of the respective nuclear-encoded CIA proteins by growth on s.d. medium for the indicated times at 30 °C. Serial tenfold dilutions were spotted onto agar plates containing either SGal (minimal medium plus galactose) or s.d. medium, and growth was continued for 2 days at 30 °C. None of the *T. hominis* genes improved growth of the yeast CIA protein-depleted cells. (**b**) *T. hominis* ThCfd1 and His-tagged ThNbp35 were co-expressed in *E. coli* and co-purified as a complex by affinity chromatography (insight, right). Fe/S clusters were chemically reconstituted on the ThCfd1–HisThNbp35 complex resulting in a dark-brown protein solution (inset, left). Ultraviolet–vis spectroscopy revealed an absorption peak around 420 nm indicative of the formation of [4Fe–4S] clusters on the complex. (**c**) The presence of [4Fe–4S] clusters was confirmed by X-band EPR spectroscopy of a reduced sample (40 μM, treated with 0.2 mM Na-dithionite) recorded at 10 K. Experimental conditions: frequency 9.6359 GHz, power 1 mW, modulation 0.75 mT/100 kHz. The numbers are the principal *g* values obtained by simulation.

without ThNbp35, only small amounts of Fe/S cluster were bound, even after reconstitution (Supplementary Fig. 8b) showing that this protein only binds clusters in a labile fashion. All these findings are strikingly similar to those made previously for yeast Cfd1–Nbp35 (ref. 35). Taken together, the results demonstrate the binding of two [4Fe–4S] clusters to ThCfd1–ThNbp35, and suggest that it can potentially serve as the microsporidian CIA scaffold complex.

Eukaryotes contain multiple monothiol glutaredoxins[36] that function in different cellular compartments, with mitochondrial yeast Grx5 and human GLRX5 playing a role in cellular Fe/S protein biogenesis[37–39], and cytosolic yeast Grx3–Grx4 and human GLRX3 (PICOT) being involved in intracellular iron distribution and cytosolic-nuclear Fe/S protein biogenesis[40–42]. We previously showed the functional complementation of a Grx5-depleted yeast cell by an *E. cuniculi* monothiol glutaredoxin (previously annotated as EcGrx5 (ref. 7)) on its targeting to mitochondria with a foreign presequence. Here, we tested the *T. hominis* Grx homologue (annotated here as ThGrx3) for yeast *GRX5* mutant complementation, but unlike EcGrx5, ThGrx3 did not rescue the growth of the Grx5-depleted yeast mutant on targeting to mitochondria (Supplementary Table 2). Using specific antibodies against ThGrx3, we immuno-detected the protein mainly in the

supernatant fraction of *T. hominis*-infected RK-13 cells after differential centrifugation (Supplementary Fig. 9a). In agreement with these data, a predominant cytosolic localization was observed by immunofluorescence microscopy using the same antibody (Supplementary Fig. 9b). Although these data cannot exclude a minor pool of ThGrx3 in mitosomes, we find no unambiguous evidence for that location. Hence, it appears that ThGrx3 is mainly cytosolic and thus is similar to cytosolic Grx3–Grx4. It has already been shown that the glutaredoxin (Grx) domain of yeast Grx3 can complement a yeast *GRX5* mutant on targeting to mitochondria[43], and this could explain why we had observed complementation by the *E. cuniculi* EcGrx5 (ref. 7). Because cytosolic monothiol glutaredoxins contain an additional N-terminal thioredoxin-like (Trx) domain[36], we searched for its presence in the microsporidian glutaredoxins. We found a divergent, N terminally-truncated Trx domain in EcGrx5 using standard bioinformatics tools and a similar (40% identity) stretch of sequence in ThGrx3 that was nevertheless not recognized as a Trx domain in the same analyses (Supplementary Figs 1c and 9c). Taken together, our data suggest a predominant cytosolic localization of ThGrx3 and, consistent with that location, some structural similarity of both microsporidian proteins to yeast Grx3/Grx4, rather than Grx5 (Supplementary Table 1).

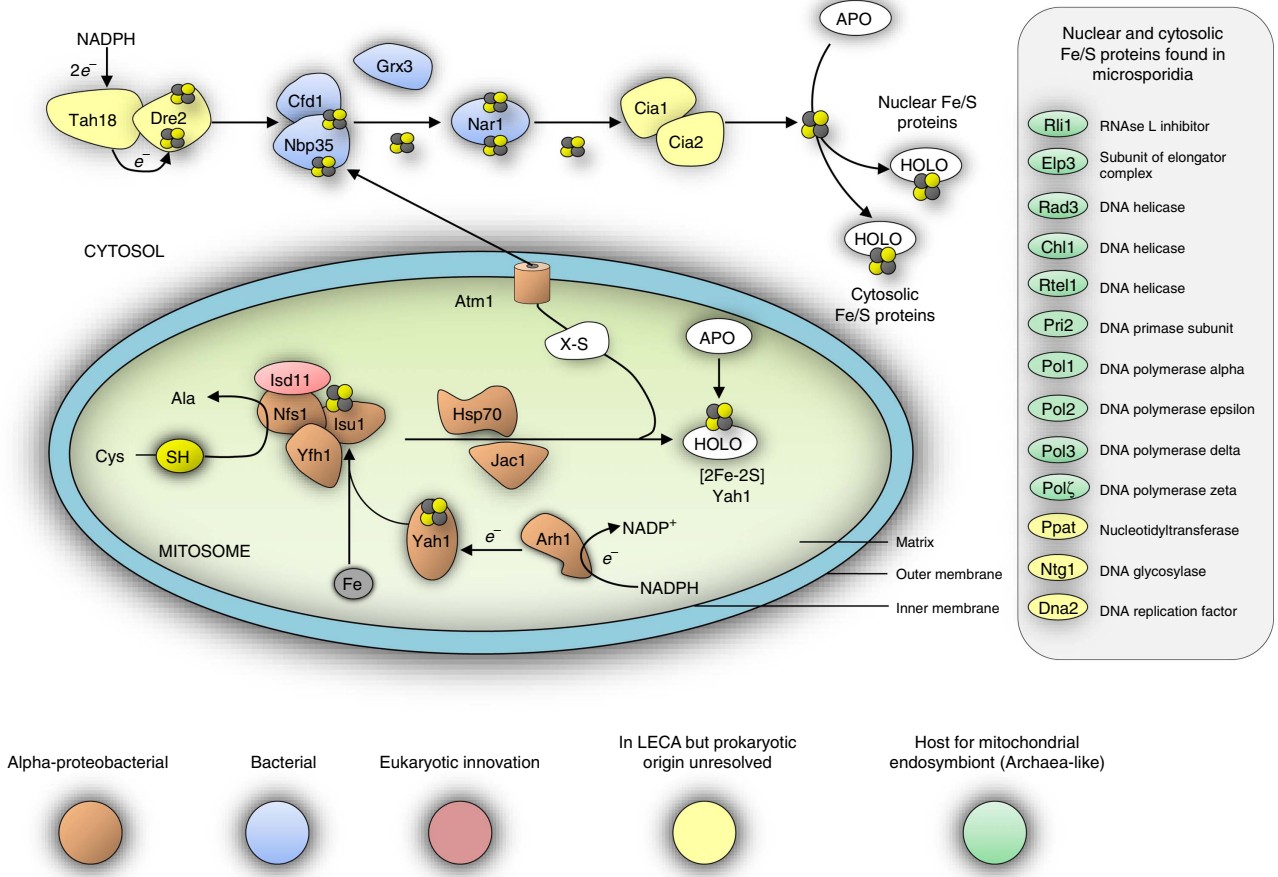

**Figure 6 | Evolutionary origins of microsporidian Fe/S-related proteins.** The cartoon shows the components of the eukaryotic ISC and CIA machineries conserved on the genomes of *E. cuniculi* and *T. hominis*. Individual components are coloured according to their inferred evolutionary origins (Supplementary Fig. 2). Components of the mitosomal ISC pathway appear to have originated from the mitochondrial endosymbiont. The CIA pathway is largely bacterial in character, and not archaeal as might be expected based on evidence for an archaeal origin of the host for the mitochondrial endosymbiosis[48,49]. By contrast, important nuclear and cytosolic Fe/S proteins do appear to have an archaeal origin. Monophyly of eukaryotic sequences, including those from microsporidians, is generally observed, suggesting that there is strong negative selection against gene replacement due to the important roles that Fe/S proteins play in eukaryotic physiology.

**Evolutionary origin of microsporidian ISC and CIA components**. We inferred phylogenies for the mitochondrial ISC and cytosolic CIA components that are conserved in the *E. cuniculi* and *T. hominis* genomes, as well as for important conserved nuclear and cytosolic Fe/S proteins involved in protein translation and in DNA replication and repair (Supplementary Fig. 2). The monophyly of eukaryotic sequences is consistent with the presence of these proteins and pathways in the last eukaryotic common ancestor (LECA), and provides evidence of the strong negative selection pressure against their subsequent replacement. Components of the core microsporidian ISC pathway and Atm1 generally formed a clade with other eukaryotes and Alphaproteobacteria, or else the tree contained an unresolved clade of eukaryotes, Alphaproteobacteria and miscellaneous prokaryotes (Supplementary Fig. 2). This suggests that the ISC pathway of microsporidians and other eukaryotes was vertically inherited from the mitochondrial endosymbiont, which is generally considered to descend from an Alphaproteobacterium (Fig. 6)[3,44,45]. The sole exception to this pattern is Isd11, which appears to be an early eukaryotic innovation[46], since we did not detect any homologues of this protein among prokaryotes. Late-acting mitochondrial ISC components (that is, Isa1, Isa2, Iba57, Nfu1, Bol3 and Ind1 (ref. 10)) specifically required for the

maturation of [4Fe–4S] proteins also appear to have originated from the mitochondrial endosymbiont, but as discussed above have subsequently been lost in microsporidia along with mitosomal [4Fe–4S] proteins (Supplementary Table 1).

The host lineage for the mitochondrial endosymbiont is now thought to have originated from within the Archaea[47–49]. Apart from the genomic presence of *SufB–SufC* genes in Archaea[50], not much is known about the proteins required to assemble archaeal Fe/S proteins. Nevertheless, the components of the cytosolic Fe/S protein assembly (CIA) pathway might be expected to have an archaeal ancestry, reflecting the fundamental requirement of the aboriginal host cell to make its own essential inventory of cytosolic (now both cytosolic and nuclear) Fe/S proteins. Contrary to this expectation, components of the eukaryotic CIA pathway, including the CIA components contained in *T. hominis* and *E. cuniculi*, do not show a specific affiliation to archaeal homologues. Cfd1, Nbp35, Nar1 and Grx3 are related to bacterial homologues, but there is no strong tree-based evidence for an origin from Alphaproteobacteria (Supplementary Fig. 2). Cia1, Cia2 and Tah18 are strongly conserved among eukaryotes suggesting their presence in the LECA, but their relationship to prokaryotic homologues was not clearly resolved (Supplementary Fig. 2). The microsporidian

Dre2 sequences are shorter than other eukaryotic Dre2 sequences and lack the N-terminal region, which has been shown to be dispensable for yeast viability[51,52]. Microsporidia have retained the conserved Dre2 C-terminal domain that interacts with Tah18 and carries both [2Fe–2S] and [4Fe–4S] clusters[52,53] (Supplementary Fig. 1i). Recent analyses of Dre2 from other eukaryotes suggested that it was present in the LECA[54], but did not resolve its specific prokaryotic origin. Taken together, these data suggest that the CIA pathway is of mainly bacterial heritage and was already in place in the LECA.

By contrast to the predominantly bacterial origin of both the ISC and CIA pathways, important nuclear and cytosolic Fe/S proteins from eukaryotes including microsporidians appear to have been inherited from the archaeal host (Fig. 6; Supplementary Fig. 2). Phylogenies for Elp3 and the paralogous genes Rad3, Chl1 and RTEL1 are consistent with a closer relationship of eukaryotic sequences to Archaea rather than Bacteria, but the low support values at this depth from trees of single proteins prevented inferences of a particular founding archaeal lineage. Homologues of the ABC protein Rli1 (full-length version including both the Fe/S and ABC domains) and the primase Pri2 were only found among eukaryotes and Archaea. On the basis of phylogeny and the close structural similarities between the B-family of replicative DNA polymerases of eukaryotes and Archaea, it has been suggested that eukaryotic proteins are derived from archaeal versions, with eukaryotic DNA polymerases of the epsilon type having a chimeric prokaryotic ancestry[55]. Our analyses for the four Fe/S-cluster-containing paralogues ($\alpha$, $\delta$, $\epsilon$, $\zeta$) of replicative B family DNA polymerases from eukaryotes including microsporidians are consistent with that hypothesis, but the support values were generally low. Trees for Ppat, Ntg1 and Dna2 sequences are consistent with the early common origin of most eukaryotic sequences, but were insufficiently resolved to infer their prokaryotic affinities (Fig. 6; Supplementary Fig. 2).

## Discussion

In microsporidians and other eukaryotes, a bacterial pathway for making Fe/S clusters now serves to support the maturation of 'archaeal-like' nuclear and cytosolic proteins. The ISC pathway components that function inside the mitochondrion/mitosome provide evidence for their origin from the alphaproteobacterial mitochondrial endosymbiont[5]. The cytosolic pathway is also largely bacterial in character, but the proteins show no particular link to the Alphaproteobacteria in our trees. Strictly speaking, our analyses cannot exclude the possibility of the co-origin of both pathways, because the genomes of modern Alpha-proteobacteria are chimeric due to lateral gene transfer events between prokaryotes[56], and there is no reason to believe that ancient Alphaproteobacteria were any different[5]. However, it is interesting that the Fe/S protein biogenesis pathway inside the organelle retains strong evidence of alphaproteobacterial ancestry but the CIA components do not. An origin of both the ISC and CIA pathway 'en bloc' would have facilitated the replacement of the original archaeal host pathways for Fe/S protein biosynthesis with an already co-functioning pathway of bacterial character.

Our data strongly suggest that the function of the *T. hominis* mitosome is to support the maturation of essential cytosolic and nuclear Fe/S proteins by a cytosolic pathway linked to the organelle-hosted ISC system by its dependence on an exported sulfur-containing substrate of unknown identity[15,16]. Interestingly, the only mitosomal Fe/S-cluster-containing protein identified in the genomes of *E. cuniculi* and *T. hominis* is the [2Fe–2S] protein ferredoxin (Yah1), which is itself an essential

component of the mitochondrial core ISC pathway representing a 'chicken and egg' situation[11]. Studies in yeast and human cells have already shown that depletion of the mitochondrial membrane potential or defective mitochondrial ISC or cytosolic CIA pathways cause genome instability and lead to induction of the DNA damage pathway[57–59]. This data demonstrates that there is strong negative selection against any perturbation of these pathways explaining why a role in Fe/S protein biogenesis is so strongly conserved—more so than ATP production for example—among diverse mitochondrial homologues[5]. The ISC and CIA pathways and target Fe/S proteins are retained in all available microsporidian genomes, and there are no other mitochondrial biosynthetic pathways similarly conserved[1,2]. A role in Fe/S protein biosynthesis therefore appears to be the essential function and remaining selective pressure for retention, of this most minimal version of the eukaryotic mitochondrion.

## Methods

**Identification of microsporidian Fe/S-related proteins.** Clusters of orthologous eukaryotic proteins (KOGs[60]) containing human or yeast proteins involved in mitochondrial or cytosolic Fe/S protein biosynthesis or nuclear or cytosolic Fe/S-cluster-containing proteins were used as the starting material for our analyses. Each KOG was supplemented with the orthologues from *T. hominis*, *Dictyostelium discoideum*, *Entamoeba histolytica* HM-1:IMSS, *Giardia lamblia* ATCC 50803, *Trichomonas vaginalis*, *Cryptosporidium parvum* Iowa type II and *Plasmodium falciparum*. These were identified using the COGNITOR method[61], which assigns a protein sequence to a KOG if the top two BLASTP hits (E-value cutoff of 0.001) from different species are members of the same KOG. In cases where a putative orthologue was not detected for one of the parasites using this approach, we also searched using HMMER[62] to apply more sensitive HMM profile (Hidden Markov model)-based methods. To identify any protein-coding genes that might have been omitted from published proteomes by mis-annotation during automated analyses we used TBLASTN to search the primary genome data.

We used HMMER3.0 to search the predicted *T. hominis* proteome using a HMM profile generated from the Atm1-like sequences of *S. cerevisiae*, *Kluyveromyces lactis*, *Lachancea thermotolerans*, *Zygosaccharomyces rouxii*, *Eremothecium cymbalariae*, *Ashbya gossypii*, *Debaryomyces hansenii*, *Candida albicans*, *Yarrowia lipolytica*, *Aspergillus niger*, *Penicillium chrysogenum*, *Magnaporthe oryzae*, *Cryptococcus neoformans*, *Schizosaccharomyces pombe*, *Danio rerio*, *Gallus gallus*, *Sus scrofa*, *Tribolium castaneum*, *Bos taurus*, *Canis lupus* and *Homo sapiens*. We identified 12 *T. hominis* sequences (at an E-value cutoff of $10^{-5}$), all of them annotated as members of the ATP-binding cassette (ABC) transporter superfamily. The highest ranking sequence was THOM_2378 (ThAtm1_1) with a single, full-length hit to the profile at an E-value of $1.7 \times 10^{-82}$. Two other sequences THOM_2384 (ThAtm1_2) and THOM_2794 (ThAtm1_3) showed significantly lower similarity, with E-values of $1 \times 10^{-44}$ and $8.1 \times 10^{-32}$, respectively. The remaining 9 *T. hominis* sequences had lower quality matches to the HMM profile.

**Evolutionary origin of microsporidian Fe/S-related proteins.** To investigate the evolutionary origins of the microsporidian Fe/S protein biogenesis machinery and nuclear and cytosolic Fe/S-cluster-containing proteins, we constructed sets of homologous sequences for each gene from eukaryotes, Bacteria and Archaea using a variety of eukaryotic and prokaryotic reference sequences as starting seeds for database searches against the NCBI non-redundant database using BLASTP. It has already been suggested that some genes of the mitochondrial Fe/S protein biosynthesis pathway originated with the mitochondrial endosymbiont[3,44,45], which is thought to have been a member of the Alphaproteobacteria. We therefore included a representative sample of Alphaproteobacteria in our analyses as well as a broad sample of bacterial and archaeal diversity. To ensure that we included the most similar prokaryotic homologues of eukaryotic proteins, we complemented our datasets with the ten best prokaryotic BLASTP hits to eukaryotic sequences.

The sequence sets resulting from this protocol were aligned with Muscle[63], Mafft[64], ProbCons[65], Kalign[66] and Fsa[67], a consensus alignment was generated using M-Coffee[68], and poorly-aligning regions were detected and removed using the 'automated1' mode in trimAl (ref. 69), which chooses from among several editing modes of varying strictness depending on the overall level of alignment conservation. We built initial 100-bootstrap maximum likelihood trees using the LG + F model in RAxML[70] to verify that our homology searches had identified orthologues of the eukaryotic genes. Bayesian phylogenetic trees were made using the C20 model implemented in PhyloBayes[71,72]. C20 is an empirical variant of the flexible CAT model[73], which contains a mixture of twenty site-specific frequency profiles estimated from a large database of alignments. Like CAT, it performs well on alignments of highly divergent sequences that are typical of

microsporidia, but it is more appropriate for small, single-gene alignments where the full CAT model may perform poorly[72]. In our Bayesian analyses, we ran two independent chains and assessed convergence by comparing the discrepancy in bipartition frequencies and a number of other model parameters (including mean posterior log likelihood, tree length and the alpha parameter for across-site rate variation) between chains. We built posterior consensus trees by pooling samples from the two chains when the maximum discrepancy in estimates for all parameters was <0.1 and the minimum sample sizes were greater than 100, as recommended by the authors (PhyloBayes manual, http://www.phylobayes.org); in the final trees, relationships with posterior support less than 0.5 were collapsed.

**Yeast mutant complementation by microsporidian genes.** The *T. hominis* genes *ThCIA1*, *ThNBP35*, *ThNAR1*, *ThCFD1*, *ThATM1_1*, *ThATM1_2 and ThATM1_3*, *ThGRX3 and ThARH1* were cloned into the yeast vector p426GPD, which contains the strong constitutive promoter of the glyceraldehyde-3-phosphate dehydrogenase gene *TDH3* (ref. 74). Targeting of *T. hominis* proteins into yeast mitochondria was achieved by cloning the proteins behind the mitochondrial presequence of *Neurospora crassa* $F_1$-ATPase subunit β[21] as previously done for ThHsp70 and ThIsu1 (ref. 7). Yeast complementation of appropriate depleted Gal-ISC or Gal-CIA cells followed established procedures[7,21].

***In vitro* reconstitution of mitosomal Fe/S-cluster synthesis.** The assay followed a procedure described earlier[11]. In brief, recombinant *T. hominis* proteins were expressed in *E. coli* with a His-tag and purified by Ni-NTA affinity chromatography followed by gel filtration (Äkta Purifier System 10, Column 16/60 Superdex 200 pg, GE Healthcare). Samples for the in vitro Fe/S-cluster synthesis assay were prepared in an anaerobic chamber (Coy Laboratory Products, Ann Arbor, MI, USA). Protein solutions and reagents were incubated under anaerobic conditions over night at 10 °C before the experiments. The 300 µl standard reaction contained 2.5 µM ThNfs1–ThIsd11, 3 µM ThYfh1, 3 µM ThYah1, 0.3 µM human FdxR and 100 µM of either ThIsu1 or CtIsu1 in buffer R (35 mM Tris–HCl, pH 8, 150 mM NaCl, 0.2 mM MgCl$_2$, 20 µM PLP, 0.5 mM NADPH, 0.5 mM sodium ascorbate, 0.3 mM FeCl$_2$). The reaction was transferred to a CD cuvette, sealed tightly and placed at 30 °C in a CD spectrophotometer equipped with an automatic stirring device (J-815, Jasco). After 2 min of temperature equilibration the Fe/S-cluster synthesis reaction was initiated by anaerobic addition of 0.5 mM cysteine. The CD signal change at 431 nm was recorded. Subsequently, full spectra were recorded from 300 to 650 nm. Initial rates were estimated by a linear fit to the initial 4.5 min of the reaction. Evaluation of the data were carried out using Origin 8 G software.

**Purification of the *T. hominis* Cfd1–Nbp35 complex.** The pETDuet-1 plasmid (Novagen) was used for heterologous co-expression of N terminally His-tagged ThNbp35 and untagged ThCfd1 in *E. coli* strain BL21 (DE3). Cells were incubated in 50 ml LB medium with ampicillin (100 mg l$^{-1}$ final concentration) at 37 °C overnight. After washing the cells were used to inoculate 2 l of TB medium and growth at 37 °C. Protein expression was induced at OD$_{600}$ of 0.8 by adding 1 mM isopropyl 1-thio-β-D-galactopyranoside in the presence of 50 µm FeCl$_3$ followed by incubation at 28 °C overnight. Cells were collected by centrifugation and shock-frozen.

Cells were resuspended in 50 ml of lysis buffer (35 mM Tris–HCl pH 7.4, 300 mM NaCl, 5% (w/v) glycerol) and disrupted by sonication (20 min on ice with 1 s intervals). After centrifugation for 90 min at 60,000g the lysate was passed over a HisTrap HP column (1 ml; GE Healthcare) and bound ThCfd1–HisThNbp35 complex eluted with an Äkta Purifier System 10 (GE Healthcare) applying an isocratic imidazole gradient from 50 mM to 1 M. The eluate was subjected to size exclusion chromatography (16/60 200 pg, GE Healthcare), and ThCfd1–HisThNbp35 was purified to apparent homogeneity in storage buffer (35 mM Tris–HCl pH 7.4, 120 mM NaCl). Purified protein was aliquoted, shock-frozen and stored at −80 °C until use.

**Chemical reconstitution of Fe/S clusters on ThCfd1–HisThNbp35.** For chemical Fe/S-cluster reconstitution ferric ammonium citrate and Li$_2$S (each 10 mM in storage buffer) were freshly prepared. ThCfd1–HisThNbp35 (80 µM) was reduced with 720 µM DTT for 1 h at 25 °C in an anaerobic chamber, and subsequently diluted with storage buffer to 20 µM final concentration. Reconstitution was started by stepwise addition of ferric ammonium citrate to a final concentration of 200 µM. Subsequently, Li$_2$S was added to a final concentration of 200 µM. After incubation for 15 min reconstituted proteins were desalted using a PD-10 column equilibrated with storage buffer and concentrated to a final concentration of 40 µM.

**EPR spectroscopy of ThCfd1–HisThNbp35.** For EPR spectroscopy, chemically reconstituted ThCfd1–HisThNbp35 complex (40 µM) was anaerobically reduced with sodium dithionite (200 µM). Samples were shock-frozen after incubation for 3 min. X-band EPR derivative spectra were recorded on a Bruker ELEXSYS E500 spectrometer equipped with a Bruker dual mode cavity (ER4116DM) and a helium flow cryostat (Oxford Instruments ESR 900). The microwave bridge was a high-sensitivity Super-X bridge (Bruker ER-049X) with integrated microwave frequency

counter. The magnetic field controller (ER032T) was calibrated with a Bruker NMR field probe (ER035M). EPR simulations were performed with the self-made routine esim_gfit (by E.B.).

**Protein expression and generation of antibodies.** Full-length *ThGRX3/5* and *ThARH1* genes were cloned using the Champion pET100 Directional TOPO Expression kit, a *TOM70* gene fragment encoding the cytoplasmic domain was cloned in the expression vector pET16b and all the proteins were expressed in BL21 (DE3) *E. coli* cells as recombinant histidine-tagged proteins. Expressed proteins were purified by gel electrophoresis for the commercial (Agrisera, Sweden) generation of rabbit antisera. Antibodies for the three candidate *T. hominis* Atm1 homologues were generated commercially (BioGenes, GmbH) against two peptides located in the most variable domains of each candidate Atm1. The sequences of the peptides were: Atm1_1: DNYIFKNMSFEIKKG and KNEKSTKNAQEMSDT; Atm1_2: RGHEETNNDGRNSN and QDDTYQNATEDRGST and Atm1_3: DTIKKLSERSPHMSK and DEGEQPAAKRYLET.

**Fractionation of infected rabbit kidney cells.** Rabbit kidney cells (RK-13) were routinely cultured in 175 cm$^2$ flasks, infected with *T. hominis* spores and grown at 37 °C in Dulbecco's Modified Eagle Medium (DMEM), containing kanamycin 100 µg ml$^{-1}$, penicillin 100 µg ml$^{-1}$, streptomycin 100 µg ml$^{-1}$ and fungizone 1 µg ml$^{-1}$ (refs 3,75). Infected cells were harvested by trypsinisation for 5 min at 37 °C followed by one wash in complete medium and two washes in PBS. A final wash was done in HSDP buffer (0.25 M sucrose, 10 mM HEPES pH 7.2) containing 1× protease inhibitor cocktail (Sigma). All subsequent manipulations were performed on ice. Cells were resuspended in cold HSDP buffer containing DNase (1 U µl$^{-1}$) and RNase (10 mg/ml) and homogenized by 50 strokes in a Dounce homogenizer or by sonication. Cell lysis was verified by trypan blue exclusion. The RK-13 cell lysate was left at room temperature for 20 min and subjected to differential ultracentrifugation in a fixed-angle Ti 70.1 rotor (Beckman) using the following parameters: 1,000g for 10 min, 10,000g for 30 min, 25,000g and 100,000g for 1 h. The final 100,000g supernatant represented the cytosolic fraction. After each centrifugation step, pellets were resuspended in PBS and protein concentration was determined using standard BCA assay. All samples were diluted in loading buffer and were boiled for 5 min except samples that were to be used for Atm1 protein visualization, which were heated at 37 °C for 10 min. Twenty micrograms of protein for each fraction was loaded per lane for SDS–PAGE and Western blot analysis using HRP-conjugated secondary antibodies (Jackson Laboratories). Proteins were visualized by chemiluminescence using a ChemiDoc XRS+ imager system (Biorad).

To affinity-purify the anti-Atm1_1 antibody, the original antiserum was diluted 1:10 in TBS-Tween 1% milk and incubated overnight with a nitrocellulose membrane blotted with recombinant Atm1_1 run on an SDS–PAGE gel. After washing the membrane three times with TBS-T 1% milk solution, the antibody was eluted with 0.1 M glycine, pH 2.5 and 0.15 M NaCl. The concentration of IgG was quantified using a BCA Protein Assay Kit (Pierce) and stored with 10% BSA at −20 °C before use for IFA and WB. As an alternative approach to reduce host background signal from the anti-Atm1_1 antiserum, it was diluted 1/250 with TBS-T 1% milk solution and incubated for 1 h (twice) with nitrocellulose membranes blotted with RK-13 total protein extract separated by SDS–PAGE. The purified antisera were then used directly for IFA and/or WB.

**Protease protection assay for mitosome-enriched fractions.** Proteinase K protection assays were performed on the 25,000g (mitosomal-enriched) pellet of infected RK-13 cells obtained as described above. The 25,000g pellet was washed twice with HSDP buffer without proteinase inhibitors and incubated with 50 µg ml$^{-1}$ proteinase K (Roche) for 20 min, or 0.2% (v/v) Triton X-100 for 10 min at room temperature followed by incubation with 50 µg ml$^{-1}$ proteinase K (Roche) for 20 min. The membrane fractions were then incubated with trichloroacetic acid at 20% (v/v) final concentration for 30 min on ice. Precipitated proteins were sedimented by centrifugation and solubilized in loading buffer. The mitosomal proteins were subsequently analysed by western blot.

**Immunofluorescence assay localization of *T. hominis* proteins.** Immunofluorescence assay (IFA) was done as described previously[7]. Briefly, RK-13 cells infected with *T. hominis* were grown on coverslips until confluent and then fixed in acetone/methanol 50:50 v/v at −20 °C for 2 h. After blocking with 5% milk in PBS, slides were incubated for 1 h with a 1% (w/v) milk/PBS solution containing the relevant antiserum, washed in PBS and then incubated for 1 h with the appropriate secondary antibodies: Rat anti-ThHsp70 was used as a mitosomal marker, the remaining antibodies against *T. hominis* proteins were raised in rabbit. Goat anti-rat or anti-rabbit antibodies (Invitrogen) conjugated to Alexa 594 (red) or 488 (green) fluorophores were used as secondary antibodies. Cells were incubated with DAPI for 5 min to visualize host cell and parasite DNA. All cells were visualized using either a Leica TCS SP2 UV confocal microscope or a Zeiss Axioimager II epifluorescence microscope with a X63 objective lens.

**Electron microscopy.** *T. hominis*-infected RK cells (RK-13) were grown as monolayers to near confluency before fixation in 0.5% glutaraldehyde in 0.2 M PIPES (buffer; pH 7.2; 15 min at room temperature). The cells were scraped from the culture flask, resuspended in 1 ml of fixative and centrifuged in a plastic Eppendorf tube (15 min at 16,000*g*). After washing three times in buffer (5 min per wash) the cell pellet was cryoprotected in 2.3 M sucrose in PBS (overnight at 4 °C). Small fragments of cell pellets were mounted onto specimen carriers, plunge-frozen in liquid nitrogen and cryo-sectioned at − 100 °C (EM FC7 ultracryomicrotome; Leica, Vienna, Austria; 80 nm thickness). Ultrathin sections were picked-up and thawed on 2.1 M sucrose and 2% w/v methylcellulose (pre-mixed in equal volumes) before mounting on carbon/pioloform-coated EM copper grids (Agar Scientific, Stansted, UK). For immunogold labelling, grids were washed in first in deionized water (three 5 min washes at 0 °C) and then in PBS (single wash at room temperature). After an initial blocking step on 0.5% fish skin gelatin (Sigma Aldrich, Poole, UK) in PBS, the grids were labelled using rabbit antisera raised against *T. hominis* ISC proteins followed by 10 nm protein-A gold (BBI Solutions, Cardiff, UK). After washes in PBS and deionized water the sections were floated on droplets of 2% w/v methylcellulose and 3% w/v uranyl acetate (mixed 9:1 before air drying (as described in ref. 76). The labelled sections were imaged with a JEOL 1200 EX transmission electron microscope operated at 80 kV on Ditabis imaging plates (DITABIS Digital Biomedical Imaging Systems AG, Pforzheim, Germany) or using a GATAN Orius 200 digital camera (GATAN, Abingdon, Oxon, UK). To quantify gold label, cell profiles were sampled systematic uniform random (SUR; ref. 77) in three individual experiments per antibody by taking 34–40 micrographs per sample at a nominal magnification of 50,000-fold. Tiff format image files were imported in Adobe Photoshop CS6 and overlaid with randomly placed square lattice grids to estimate the areas of compartments of interest (mitosomes, cytoplasm and nucleus of vegetative and sporulating cell stages of *T. hominis*) by point counting (grid spacing 15.5 nm for mitosomes, 206.1 nm for cytosol and nucleus). The estimated areas were then related to the number of gold particles to calculate the density of gold per $\mu m^2$.

To investigate the sub-organellar localization of ISC proteins we compared a random point distribution over the positively labelled mitosome profiles with the distribution of the gold label of the same mitosome profiles for each protein ($n = 37$–79 gold particles and equal number of random points per ISC protein). The distance from the centre of each gold particle was measured to the inner aspect of the inner mitosomal membrane and compared with the distance measured for the random points to the same membrane. The measurements were categorized into separate distance groups and the frequency distribution for each Fe/S-cluster protein was estimated. To evaluate the distribution of labelling over the mitosomal matrix statistically the distribution of random points and labelling were compared using the Kolmogorov–Smirnov, Mann–Whitney and Chi-squared tests.

Vegetative stages of *T. hominis* (meronts) could be identified as single or multinucleated cell profiles situated within parasite vacuoles of RK cells. Sporulating parasites (sporonts, sporoblasts or spores) were distinguished by the presence of a discernible cell wall and/or the formation of the polar tube. Cytoplasm was defined as any area enclosed by the plasma membrane excluding nuclear and mitosomal profiles. The polar tube and associated structures (lamellar polaroplast) were not included in the quantification. Nuclear profiles were defined by nucleoplasm bounded by and including the nuclear envelope. Mitosomes were identified as double membrane bound organelles measuring between 47 to 119 nm and 78 to 267 nm for minor and major axes, respectively.

**Data availability.** The authors declare that all data supporting the findings of this study are available within the article and its Supplementary Information Files or from the corresponding author upon request.

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

## Acknowledgements

We gratefully acknowledge the contribution of the Core Facility Protein Spectroscopy and Protein Biochemistry of Philipps-Universität Marburg. Technical support from Ekaterina Kozhevnikova is gratefully acknowledged. This work was supported by Marie Curie Postdoctoral Fellowships to T.A.W., E.H. and S.L., a European Research Council Advanced Investigator Grant (ERC-2010-AdG-268701) to T.M.E., and a Wellcome Trust Programme Grant (Number 045404) to T.M.E. and J.M.L. R.L. acknowledges generous financial support from Deutsche Forschungsgemeinschaft (SFB 593, SFB 987, GRK 1216, LI 415/5), LOEWE program of state Hessen, Max-Planck Gesellschaft, von Behring-Röntgen Stiftung.

## Authors contributions

J.M.L., T.M.E. and R.L. designed the study. S.-A.F., A.V.G., C.H., S.M., P.D., T.A.W., S.N., S.L., K.S., E.B., E.H. and R.P.H. performed the experiments. All authors analysed data. T.M.E. and R.L. wrote the manuscript with contributions from S.-A.F., A.V.G. and J.M.L. All authors edited and approved the final manuscript.

## Additional information

**Competing financial interests**: The authors declare no competing financial interests.

