## [Peer Review File · Nature Communications]

Reviewers' comments:

Reviewer #1 (Remarks to the Author):

This is a very strong, interesting and important paper that provides a complete functional and evolutionary picture of FeS cluster assembly in the *T. hominis* microsporidian system. The findings set a new technical standard for the study of this pathway in eukaryotes and underscore the chimaeric, endosymbiotic nature of the eukaryotic cell. First they found all the genes for 2Fe2S cluster assembly, with 4Fe4S not being needed by the parasite (interestingly). Finding these genes is a technical challenge in itself, but they localized the products and reconstituted the system in vitro (a huge effort, raises the bar for future studies) in addition to characterizing the evolutionary history of each of the components (and the substrates of FeS incorporation). The paper was submitted by internationally leading experts in their fields and Nature Comm can probably be happy to have received this submission.

I have very little to criticize about this fine piece of work, I only stumbled (got to thinking about issues) at a few places.

1. Why is Schaedler in JBC not mentioned? Nonspecialists have become aware of her evidence for glutathione sulfur species (trisulfide, persulfide, polysulfide) as a possible export substrate, the authors cite some of the many RL citations here to back the statement that substrate is unknown, but is there something wrong with the Balk JBC paper? It was getting Curr Opin stars in 2014. Maybe just mention Schaedler but that there are dissenting views at it relates to the substrate there or for other systems. What is the oxidation state of Fe at lower left in Fig. 3?

2. The evolutionary discussion is good, except maybe playing up the role of the *Monocercomonoides* paper at the crescendo of the manuscript, right before the closing two sentences (where the most important findings should be summarized). The LGT detour detracts from the main message here, also the causality implied ("has indeed allowed"). There have been other papers with earlier claims about LGT in FeS cluster assembly (*Mastigamoeba*, *Entamoeba*). Are those claims wrong? Why are they not discussed? The present paper is not an LGT paper until that sentence, and then the reader has an LGT take home message. My feeling would be keep the discussion focussed on the microsporidians, not unusual genes found (and not found) in other genomes. Have people looked in *Monocercomonoides* data as hard as the authors looked here? The rock-hard conservation of the pathway in microsporidians is the story here, not pathway replacement in a draft genome assembly.

3. The uncertain origins and sampling issues caught my attention. Degli-Esposti (Biol Direct 2016) accessed some metagenomic data recently and suddenly eukaryotic Fe-Fe hydrogenases, which were never alphaproteobacterial, are suddenly intensely alphaproteobacterial, based on newer sampling. I am NOT suggesting that the authors do more sampling or mention Hyd evolution, I am just saying that if I understand what Ku et al. are saying (better data in Ku's Nature paper), then even branching with

alphaproteobacteria is also not "strong tree-based evidence for an origin from Alphaproteobacteria " (p. 9), because the prokaryotic genomes from which the eukaryotic genes were sampled at mitochondrial origin were in flux then and have been ever since. Single origin (monophyly) in eukaryote common ancestor (where mitochondria existed) might be a stronger connection to mitochondria than some gene distributions we see in genomes (prokaryote LGT) and some of the branches we see in trees (variable sites). There is nothing in this comment to address in revision, it is just one of many thoughts that came to mind while reading this interesting paper. Readers are going to like it a lot.

Reviewer #2 (Remarks to the Author):

A. Summary of the key results

The paper by Molik and co-workers describes an impressive functional and evolutionary study of the microsporidian (*T. hominis*) ISC assembly pathways. It is demonstrated that in *T. hominis* ISC assembly occurs in the mitosomes, and the CIA was also identified and functionally verified *in vitro*. These findings hold relevance regarding the origin and early evolution of eukaryotes, underscoring their chimeric archaeal-bacterial ancestry.

B. Originality and interest: if not novel, please give references

The study by Molik and co-workers is highly original and of significant interest given that the functional biology of microsporidians, despite being of economic and medical importance, is limited due to the inability to culture and genetically manipulate microsporidian lineages. Microsporidians such as *T. hominis* hold evolutionary clues to mitochondrial evolution and essentiality, and have the potential to shed light on the origin and early evolution of eukaryotes – which makes the study by Malik et al of general interest.

C. Data & methodology: validity of approach, quality of data, quality of presentation

The methods that have been employed in this study are well-suited and regarded state-of-the-art.

D. Appropriate use of statistics and treatment of uncertainties

This seems to be in order.

E. Conclusions: robustness, validity, reliability

The conclusions that are drawn in the paper are generally robust and fully backed up by the data presented.

F. Suggested improvements: experiments, data for possible revision

The paper describes the results of an impressive amount of lab experiments as well as bioinformatics (phylogenetics) analyses, which is convincingly presented in main figures, as

well as multiple Extended and Supplementary figures and tables. I only have some minor suggested improvements (see listed below), but these should by no means delay the publication of the current manuscript.

G. References: appropriate credit to previous work?

These are appropriate.

H. Clarity and context: lucidity of abstract/summary, appropriateness of abstract, introduction and conclusions

The manuscript by Molik et al is concise, well-written, and nicely balanced. I have no further remarks about this.

Specific comments:

Page 3, 2nd paragraph: "Phylogenetic analyses recovered ThArh1 and EcArh1...": use 'orthologs' rather than 'homologs' here?

Page 4, line 1: "ISC proteins have been localized to..." -> "ISC proteins have been found to be localized at...".

Page 4, line 5: ""we used new antibodies": please explain the relevance for using new antibodies here, or rephrase.

Page 4, line 19: Explain "25 k pellet fraction" and refer to methods for details.

Page 5, line 8: "There are many potential reasons why...": Is there any indication that the T.ho proteins were correctly targeted to the yeast mitochondria?

Page 7, 1st paragraph: Was correct targeting confirmed for the T.ho Atm1 homologs?

Page 7, 2nd paragraph: explain abbreviation 'CIA'.

Page 9, 2nd paragraph: "Contrary to this expectation... (Extended Data Table 1)": As far as I can see, this observation cannot be made from the data presented in Extended Data Table 1.

Page 10, 1st paragraph: Regarding low support values, perhaps mention that this is to be expected due to weak evolutionary signals in single protein sequences.

Reviewer #3 (Remarks to the Author):

The paper claims to identify the primary mitochondrial and CIA Fe-S cluster biogenesis machinery in microsporidians. The results presented largely support these claims. The analysis of Fe-S cluster biogenesis in this obligate parasite is of broad interest to the field.

The paper has several strengths:

1. The biochemical data showing in vitro reconstitution of the ISC components clearly shows they can work together to generate a [2Fe-2S] cluster on Isu1. It would be nice to know the final cluster content of Isu1 (iron/sulfide:protein ratios) after the experiment for both control and experimental samples but this is a minor issue.

2. The authors present a number of fairly convincing sub-cellular localization experiments that establish the locations of most of the ISC and CIA components in the microsporida. In fact, it would be better if more of these figures were moved to the main body of the paper rather than as Extended Data. One other note here, Extended Data Figure 3 is more convincing than Figure 1 for establishing the subcellular localization of the ISC pathway. I would suggest swapping those two figures (making the EM experiment part of the Extended Data).

However, there were also a number of problems with the paper. The authors extrapolate quite a bit to assign functions to the various proteins based on localization data and homology despite the fact that most of their complementation studies did not succeed in the yeast mutant strains. Unlike the ISC system, they lack convincing biochemical data for atm1 and most of the CIA machinery. This type of extrapolation is premature without further data. Also, most of the extensive phylogenetic analysis does not fit well with the experimental portion of the paper. Phylogenetic results are presented as observations but do not really inform or guide any actual experiments in the paper. The large number of phylogenetic comparisons would fit better in a review article than in a primary research article (unless experiments were presented that test some of the hypotheses generated by the phylogenetic analysis). This reviewer would suggest greatly condensing the phylogenetic analysis and focusing on the core experimental results within the paper.

It is clear that the majority of the mitochondrial ISC homologues do not complement deletion mutations of their yeast counterparts. However, the mitochondrial system also has homologues in the bacteria. It would be worthwhile to attempt complementation studies in a bacterium such as *E. coli* that is well-studied with clear ISC phenotypes that could be tested. The authors are already over-expressing their proteins in *E. coli* so it seems there would be no major issues with expression levels of the recombinant proteins. Perhaps expressing the entire ISC machinery from a Duet or similar vector system could work to complement the bacterial phenotypes.

Extended Data Figure 9C needs more experimental details. What buffer was used for the reconstitution? How were iron and sulfide provided? Was the sample purified after reconstitution to remove iron-sulfide species (including potentially Fe-S-DTT species that can show absorbance similar to [4Fe-4S] clusters)? In general UV-visible absorption spectroscopy is not sufficient to assign a cluster type. Fe:S:protein ratios should be reported and/or other techniques such as EPR or Mossbauer spectroscopy should be used to confirm the cluster type and amount.

Manuscript NCOMMS-16-18621-T (Freibert *et al.*)

Response to Reviewers' comments:

(our comments are in red)

We thank the reviewers for their constructive comments. We have used them to substantially improve our manuscript by addressing all of their comments (see below for details) and by adding new experimental data as requested (this resulted in a change of authorship order and in the addition of a new co-author who helped with the EPR spectroscopic analyses). To adhere to the format of *Nature Communications* we have expanded the introductory section and added subheadings in a Results and Discussion section. As suggested by one of the reviewers we have moved three Supplementary Figures to the main text. With these alterations, we hope that our manuscript is now ready for publication in *Nature Communications*.

Reviewer #1 (Remarks to the Author):

This reviewer was very supportive and made a couple of suggestions for minor changes to improve our MS that we have taken on board.

This is a very strong, interesting and important paper that provides a complete functional and evolutionary picture of FeS cluster assembly in the *T. hominis* microsporidian system. The findings set a new technical standard for the study of this pathway in eukaryotes and underscore the chimaeric, endosymbiotic nature of the eukaryotic cell. First they found all the genes for 2Fe2S cluster assembly, with 4Fe4S not being needed by the parasite (interestingly). Finding these genes is a technical challenge in itself, but they localized the products and reconstituted the system in vitro (a huge effort, raises the bar for future studies) in addition to characterizing the evolutionary history of each of the components (and the substrates of FeS incorporation). The paper was submitted by internationally leading experts in their fields and Nature Comm can probably be happy to have received this submission.

I have very little to criticize about this fine piece of work, I only stumbled (got to thinking about issues) at a few places.

1. Why is Schaedler in JBC not mentioned? Nonspecialists have become aware of her evidence for glutathione sulfur species (trisulfide, persulfide, polysulfide) as a possible export substrate, the authors cite some of the many RL citations here to back the statement that substrate is unknown, but is there something wrong with the Balk JBC paper? It was getting Curr Opin stars in 2014. Maybe just mention Schaedler but that there are dissenting views at it relates to the substrate there or for other systems. What is the oxidation state of Fe at lower left in Fig. 3?

The interesting paper by Schaedler *et al.* (JBC 2014) suggested that GSSSG or related glutathione persulfides may be exported by Atm1 to the cytosol, based on in vitro studies. However, other papers published shortly afterwards have questioned this suggestion. For instance, two independent papers (Ida *et al.*, PNAS 111, 7606–7611; Libiad *et al.*, JBC 289,30901–30910) showed that GSSH (the precursor of the putative Atm1 substrates) can be produced in both mitochondria and cytosol. This makes the transport of such a compound appear to be unnecessary. These papers and the still

unclear nature of the Atm1 substrate have been discussed in depth in a recent review (Lill et al., Eur. J. Cell Biol 2015) which we now cite to meet the reviewer's concern.

2. The evolutionary discussion is good, except maybe playing up the role of the *Monocercomonoides* paper at the crescendo of the manuscript, right before the closing two sentences (where the most important findings should be summarized). The LGT detour detracts from the main message here, also the causality implied ("has indeed allowed"). There have been other papers with earlier claims about LGT in FeS cluster assembly (*Mastigamoeba*, *Entamoeba*). Are those claims wrong? Why are they not discussed? The present paper is not an LGT paper until that sentence, and then the reader has an LGT take home message. My feeling would be keep the discussion focussed on the microsporidians, not unusual genes found (and not found) in other genomes. Have people looked in *Monocercomonoides* data as hard as the authors looked here? The rock-hard conservation of the pathway in microsporidians is the story here, not pathway replacement in a draft genome assembly.

We have taken the reviewers advice and removed the reference to the *Monocercomonoides* data. We now focus the last paragraph on our own data and its implications and importance.

3. The uncertain origins and sampling issues caught my attention. Degli-Esposti (Biol Direct 2016) accessed some metagenomic data recently and suddenly eukaryotic Fe-Fe hydrogenases, which were never alphaproteobacterial, are suddenly intensely alphaproteobacterial, based on newer sampling. I am NOT suggesting that the authors do more sampling or mention Hyd evolution, I am just saying that if I understand what Ku et al. are saying (better data in Ku's Nature paper), then even branching with alphaproteobacteria is also not "strong tree-based evidence for an origin from Alphaproteobacteria " (p. 9), because the prokaryotic genomes from which the eukaryotic genes were sampled at mitochondrial origin were in flux then and have been ever since. Single origin (monophyly) in eukaryote common ancestor (where mitochondria existed) might be a stronger connection to mitochondria than some gene distributions we see in genomes (prokaryote LGT) and some of the branches we see in trees (variable sites). There is nothing in this comment to address in revision, it is just one of many thoughts that came to mind while reading this interesting paper. Readers are going to like it a lot.

We agree with the reviewer that LGT is a thorny issue deserving of more discussion than we have space for in our present MS. Although the reviewer did not request that we revise our text we have modified it slightly to clarify our meaning. In addition, we have cited Embley and Martin 2006 (original citation 6) to supplement the existing Ku et al. citation. This paper discussed the difficulties of inferring gene origins from the mitochondrial endosymbiont in the light of extensive prokaryote-to-prokaryote LGT.

Reviewer #2 (Remarks to the Author):

This reviewer was very supportive and made some suggestions for minor corrections/clarifications, all of which we have addressed.

A. Summary of the key results

The paper by Molik and co-workers describes a impressive functional and evolutionary study of the microsporidian (*T. hominis*) ISC assembly pathways. It is demonstrated that in *T. hominis* ISC assembly occurs in the mitosomes, and the CIA was also identified and functionally verified in vitro. These findings hold relevance regarding the origin and early evolution of eukaryotes, underscoring their chimeric archaeal-bacterial ancestry.

B. Originality and interest: if not novel, please give references

The study by Molik and co-workers is highly original and of significant interest given that the functional biology of microsporidians, despite being of economic and medical importance, is limited due to the inability to culture and genetically manipulate microsporidian lineages. Microsporidians such as *T. hominis* hold evolutionary clues to mitochondrial evolution and essentiality, and have the potential to shed light in the origin and early evolution of eukaryotes – which makes the study by Malik et al of general interest.

C. Data & methodology: validity of approach, quality of data, quality of presentation

The methods that have been employed in this study are well-suited and regarded state-of-the-art.

D. Appropriate use of statistics and treatment of uncertainties

This seems to be in order.

E. Conclusions: robustness, validity, reliability

The conclusions that are drawn in the paper are generally robust and fully backed up by the data presented.

F. Suggested improvements: experiments, data for possible revision

The paper describes the results of an impressive amount of lab experiments as well as bioinformatics (phylogenetics) analyses, which is convincingly presented in main figures, as well as multiple Extended and Supplementary figures and tables. I only have some minor suggested improvements (see listed below), but these should by no means delay the publication of the current manuscript.

G. References: appropriate credit to previous work?

These are appropriate.

H. Clarity and context: lucidity of abstract/summary, appropriateness of abstract, introduction and conclusions

The manuscript by Molik et al is concise, well-written, and nicely balanced. I have no further remarks about this.

Specific comments:

Page 3, 2nd paragraph: “Phylogenetic analyses recovered ThArh1 and EcArh1...”: use ‘orthologs’ rather than ‘homologs’ here?

Done, we now use “orthologues” in this case.

Page 4, line 1: “ISC proteins have been localized to...” -> “ISC proteins have been found to be localized at...”.

Done, we now write “shown to localize”

Page 4, line 5: “we used new antibodies”: please explain the relevance for using new antibodies here, or rephrase.

Rephrased to “we made new antibodies to *T. hominis* Thlsu1 and ThYfh1 for immunoelectron microscopy”.

Page 4, line 19: Explain “25 k pellet fraction” and refer to methods for details.

Done, we now write “were accumulated in the 25,000 xg (mitosome-enriched) pellet of infected RK-13 cells, which also contained the *T. hominis* homologue of mitochondrial Hsp70 (ThHsp70), a validated mitochondrial marker protein...”. We have also modified the respective figure legends.

Page 5, line 8: “There are many potential reasons why...”: Is there any indication that the *T.ho* proteins were correctly targeted to the yeast mitochondria?

Yes. Whenever possible (antibodies available), we did immunoblots. Moreover, from our long-standing experience on mitochondrial protein import, we know that the F1beta presequence is highly reliable in faithfully targeting attached proteins to the mitochondrial matrix. (Gerber et al. 2004 citation 22). We have previously shown that fungal presequences effectively guide import of microsporidian proteins into yeast mitochondria (e.g., Western blots in Supplemental Figure S8 in Goldberg et al. 2008 for *E. cuniculi* EcYfh1 and EcGrx3/5) and complementation of yeast mutants in the same paper (e.g., EcYfh1 and EcGrx3/5 in Figure S8) and for *T. hominis* Thlsu1 (Goldberg et al. 2008) and Thlsd11 in the present manuscript.

Page 7, 1st paragraph: Was correct targeting confirmed for the *T.ho* Atm1 homologs?

We used the above mentioned presequence for targeting of the putative Atm1 homologues and we now make this clear in the text: “We tested all three *T. hominis* Atm1 candidates for complementation of a yeast ATM1 mutant using high and low level expression vectors with the addition of a fungal mitochondrial presequence at the N terminus of the *T. hominis* sequences...”.

It is possible that correct membrane assembly of the microsporidian ABC transporter may have failed in the foreign environment, and this could be one reason for the lack of complementation.

Page 7, 2nd paragraph: explain abbreviation ‘CIA’.

Done. We now briefly explain the cytosolic iron-sulfur protein assembly (CIA) pathway in the Introduction.

Page 9, 2nd paragraph: “Contrary to this expectation... (Extended Data Table 1)”: As far as I can see, this observation cannot be made from the data presented in Extended Data Table 1.

Our thanks to the reviewer for spotting the unnecessary reference to Extended Data Table 1 at this point in our MS. We have now removed it.

Page 10, 1st paragraph: Regarding low support values, perhaps mention that this is to be expected due to weak evolutionary signals in single protein sequences.

Done, we now write “but the low support values at this depth from trees of single proteins prevented inferences of a particular founding archaeal lineage.”

Reviewer #3 (Remarks to the Author):

This reviewer was supportive but had some issues for us to resolve which we think were well-taken. We have addressed all these points by new experiments and better explanations.

The paper claims to identify the primary mitochondrial and CIA Fe-S cluster biogenesis machinery in microsporidians. The results presented largely support these claims. The analysis of Fe-S cluster biogenesis in this obligate parasite is of broad interest to the field.

The paper has several strengths:

1. The biochemical data showing in vitro reconstitution of the ISC components clearly shows they can work together to generate a [2Fe-2S] cluster on Isu1. It would be nice to know the final cluster content of Isu1 (iron/sulfide:protein ratios) after the experiment for both control and experimental samples but this is a minor issue.

In our Nat. Commun paper of 2014 (Webert et al.) we have shown that our ISC-driven reconstitution system can assemble about one [2Fe-2S] cluster per Isu1 dimer. This number was based on a comparison of CD spectra ($\Delta\epsilon$ at 431 nm) of chemically reconstituted (where the Fe and S content was directly measured) and ISC-reconstituted Isu1. In the latter case, a direct measure of the cluster content on Isu1 is more demanding and error-prone, because of the presence of iron and the [2Fe-2S] ferredoxin Yah1 in the reaction mixture. The comparison with the yeast, *Chaetomium* (or human) systems clearly shows that the ISC-driven reconstitution of ThIsu1 also generates roughly one [2Fe-2S] cluster per ThIsu1 dimer. We now have included details on the quantitation of the Fe/S content of ISC-reconstituted ThIsu1 in the Results (Fig. 3).

2. The authors present a number of fairly convincing sub-cellular localization experiments that establish the locations of most of the ISC and CIA components in the microsporidia. In fact, it would be better if more of these figures were moved to the main body of the paper rather than as Extended Data. One other note here, Extended Data Figure 3 is more convincing than Figure 1 for establishing the subcellular localization of the ISC pathway. I would suggest swapping those two figures (making the EM experiment part of the Extended Data).

The two data sets provide complementary information on the location of the ISC

components and together they make a convincing case. We would prefer to keep the EM data as a main figure but we do agree with the reviewer that extended data figure 3 should also be included in the main text (now as the new Fig. 2). We also have moved the Atm1 localization data (former Extended Fig. 9) to the main part (new Fig. 5).

However, there were also a number of problems with the paper. The authors extrapolate quite a bit to assign functions to the various proteins based on localization data and homology despite the fact that most of their complementation studies did not succeed in the yeast mutant strains. Unlike the ISC system, they lack convincing biochemical data for atm1 and most of the CIA machinery. This type of extrapolation is premature without further data.

Based upon i) the retention of Atm1 and CIA homologues in the genomes of highly reduced microsporidian fungal parasites, ii) the common ancestry of these proteins with functionally characterized Atm1 and CIA proteins from yeast, and iii) the general conservation of key residues, we believe it is reasonable to infer similar functions for the yeast and microsporidian proteins, as a working hypothesis. Our paper clearly states where we lack functional (experimental) validation. We are therefore confident that there is no attempt to mislead the reader and that we have been suitably cautious in our interpretations. For example, we provide compelling evidence that a putative Atm1 localizes to *T. hominis* mitosomes (new Fig. 5), but we also made it clear that we were unable to functionally confirm that ThAtm1_1 is a true mitochondrial orthologue of yeast Atm1 (see lines 8-10, page 7).

Also, most of the extensive phylogenetic analysis does not fit well with the experimental portion of the paper. Phylogenetic results are presented as observations but do not really inform or guide any actual experiments in the paper. The large number of phylogenetic comparisons would fit better in a review article than in a primary research article (unless experiments were presented that test some of the hypotheses generated by the phylogenetic analysis). This reviewer would suggest greatly condensing the phylogenetic analysis and focusing on the core experimental results within the paper.

We respectfully disagree that the phylogenetic analyses do not fit well with the experimental portion of the paper. In our opinion (and in full agreement with reviewers 1 and 2) the combination of the two approaches is a great strength of our work, and provides additional generality to our experiments. The experiments were first motivated by the evolutionary observation that Fe/S cluster biosynthesis is the only mitochondrial biosynthetic function that has been retained in the highly reduced genomes of Microsporidia and other parasitic protists. Importantly, it is the phylogenetic analyses that demonstrate the (unexpected) different ancestries of the pathways and substrate proteins and provide the evidence for ancient chimerism at the origin of eukaryotes. We therefore appreciate the view of reviewers 1 and 2 that this combination of experimental data and evolutionary biology is a major strength of our paper. The balance between the two approaches also fits the brief of *Nature Communications* to be a multidisciplinary journal catering for a broad readership.

It is clear that the majority of the mitochondrial ISC homologues do not complement deletion mutations of their yeast counterparts. However, the mitochondrial system also

has homologues in the bacteria. It would be worthwhile to attempt complementation studies in a bacterium such as *E. coli* that is well-studied with clear ISC phenotypes that could be tested. The authors are already over-expressing their proteins in *E. coli* so it seems there would be no major issues with expression levels of the recombinant proteins. Perhaps expressing the entire ISC machinery from a Duet or similar vector system could work to complement the bacterial phenotypes.

Microsporidia are fungi and we used yeast because it is the most closely related and best-studied eukaryotic model system for the complementation experiments using microsporidian ISC homologues. In parallel, we successfully pursued an *in vitro* experimental approach to test our hypothesis that the microsporidian mitochondrial ISC pathway is functional. As this referee comments at the beginning of his/her review “the biochemical data showing *in vitro* reconstitution of the ISC components clearly shows they can work together to generate a [2Fe-2S] cluster on Isu1”. This is consistent with our previous demonstration (Goldberg et al. 2008) that *T. hominis* Isu1 and *E. cuniculi* Yfh1 and Grx5 are functional homologues of *S. cerevisiae* as shown by complementation. It is therefore not clear to us what new critical insights would accrue from repeating the complementation experiments in *E. coli*, and we would rather not embark on such a substantial body of new work (involving the genetic generation of *E. coli* depletion mutants). As the reviewer will appreciate there are also no guarantees that a distantly related prokaryotic system (that is evolutionary far more distant from microsporidia than yeast) would necessarily behave any better than yeast in such experiments.

Extended Data Figure 9C needs more experimental details. What buffer was used for the reconstitution? How were iron and sulfide provided? Was the sample purified after reconstitution to remove iron-sulfide species (including potentially Fe-S-DTT species that can show absorbance similar to [4Fe-4S] clusters)? In general UV-visible absorption spectroscopy is not sufficient to assign a cluster type. Fe:S:protein ratios should be reported and/or other techniques such as EPR or Mossbauer spectroscopy should be used to confirm the cluster type and amount.

We agree with the reviewer that the data presented in Extended Data Fig. 9C was incomplete. We therefore have performed further biochemical and spectroscopic analyses on the holo-ThCfd1-ThNbp35 complex which we now present as **new Figure 5** in the main text (plus previous data now shown as Supplementary Fig. 8). In brief, we have added experimental details of the reconstitution procedure (Methods), determined the Fe and S content by chemical analysis (mentioned in Results), and recorded EPR spectra as suggested (one representative spectrum shown in new Fig. 5c). Together, the results clearly demonstrate the presence of two [4Fe-4S] clusters on the ThCfd1-ThNbp35 dimer. Based on the striking similarity of the *T. hominis* complex to the corresponding yeast protein complex (Netz et al., JBC 2012), we propose that ThCfd1-ThNbp35 serves as the functional microsporidian CIA scaffold complex.

REVIEWERS' COMMENTS:

Reviewer #1 (Remarks to the Author):

The authors have done a fine job with the revisions, dealing in direct and convincing manner with the comments by all referees, even adding new experimental data. This is a very strong paper with a very important message and should go forward to production and publication without delay.

Reviewer #3 (Remarks to the Author):

This reviewer is satisfied with the changes submitted in the revised manuscript.